# Anti-melanogenic Activity of Auraptene via ERK-mediated MITF Downregulation

**Min-Jin Kim [1], Sang Suk Kim [2], Kyung-Jin Park [2], Hyun Joo An [2], Young Hun Choi [2], Nam Ho Lee [1] and Chang-Gu Hyun [1,*]**

[1]  Cosmetic Sciences Center, Department of Chemistry and Cosmetics, Jeju National University, Jeju 63243, Korea; alswls0428@nate.com (M.-J.K.); namho@jejunu.ac.kr (N.H.L.)
[2]  Citrus Research Institute, National Institute of Horticulture and Herbal Science, RDA, Seogwipo 63607, Korea; sskim0626@korea.kr (S.S.K.); pkj5690@korea.kr (K.-J.P.); hjan67@korea.kr (H.J.A.); yhunchoi@korea.kr (Y.H.C.)
*  Correspondence: cghyun@jejunu.ac.kr; Tel.: +82-64-754-3542

**Abstract:** Auraptene is the most abundant naturally occurring geranyloxycoumarin. It is primarily isolated from plants belonging to the Rutaceae family, many of which, such as citrus fruits, are used as food in many countries. Auraptene is a biologically active secondary metabolite that possesses valuable properties. The aim of this study was to investigate the in vitro inhibitory effects of auraptene on melanogenesis and the enzymes associated with it, such as tyrosinase, tyrosinase-related protein (TRP)-1, and TRP-2, in B16F10 murine melanoma cells. We found that auraptene significantly attenuated melanin synthesis and reduced the activity of intracellular tyrosinase, which was the rate-limiting melanogenic enzyme. Western blotting analysis showed that auraptene decreased tyrosinase and TRP-2 protein expression. In addition, auraptene significantly decreased the expression of microphthalmia-associated transcription factor (MITF), a key regulator of melanogenesis. Extracellular signal-regulated kinase (ERK) activation has been reported to be involved in the inhibition of melanogenesis. Thus, we next investigated if the hypopigmentary effects of auraptene were related to the activation of ERK. Auraptene was found to induce phosphorylation of ERK in a dose-dependent manner. Our results suggest that auraptene inhibits melanogenesis by activating the ERK pathway-mediated suppression of MITF and its downstream target genes, including tyrosinase. Therefore, auraptene may be used as a whitening agent in the development of functional cosmetics.

**Keywords:** B16F10 melanoma cell; auraptene; melanin; tyrosinase; MITF; whitening

## 1. Introduction

Destruction of the ozone layer is accelerating due to severe air pollution. A depleted ozone layer would allow more ultraviolet (UV) rays to reach us. Regardless of sex, it is important for us to protect our skin from excessive exposure to UV rays. Many cosmetic companies are interested in using natural products in their research to develop functional cosmetics for skin protection from UV rays [1]. Human skin is sensitive to UV rays that generate reactive oxygen species, which cause damage to skin cells. The formation of pigment in the skin is caused by an increase in melanin production by internal and external factors such as UV rays and signaling substances [2,3].

Melanin is the pigment responsible for the characteristic color of the skin, hair, and eyes. It plays an important protective role against UV light-induced skin damage [4]. Melanin is secreted by melanocyte cells in the basal layer of the epidermis [5]. The melanocytes may produce excessive melanin (melasma) or result in other hyperpigmentation diseases under chronic sun exposure [6,7]. Thus, the inhibition of melanin production has been the focus of medical and cosmetic treatments for skin depigmentation and

lightening. Melanin is biosynthesized in a complex process called melanogenesis, which involves three main melanogenic enzymes: tyrosinase, tyrosinase-related protein-1 (TRP-1), and TRP-2. Tyrosinase is the key enzyme in the first two steps of melanin biosynthesis [8–10]. During the process, tyrosinase converts L-tyrosine to dihydroxyphenylalanine (DOPA) through hydroxylation, and L-DOPA is oxidized to dopa-quinoneandindole-5,6-quinone, which is a substrate for melanin synthesis [11].

Currently, ascorbic acid, kojic acid, hydroquinone, benzoic acid, retinoids, and arbutin are commonly used as tyrosinase inhibitors. Despite having strong whitening effects, the use of kojic acid and arbutin is limited by product safety and economic efficiency [12]. For this reason, the search for safer and effective skin depigmentation agents continues in the field of cosmetics.

A variety of plant molecules provide an interesting and largely unexplored source of potential skin-care cosmetics. They are normally synthesized for a specific protective function in plant, and can also act on human cells, modifying their metabolism. Therefore, the present study focused on whether or not auraptene inhibits melanogenic protein expression in mouse melanoma cells [13–15].

Auraptene is a natural bioactive monoterpene coumarin ether. It was first isolated from members of the genus Citrus. Auraptene has shown a remarkable effect in the prevention of degenerative diseases. Many studies have reported the effect of auraptene as a chemopreventive agent against various cancers—such as liver, skin, tongue, esophagus, and colon cancers—in rodent models [16]. However, the effects of auraptene on skin-related cells, especially the molecular aspects of its anti-melanogenesis effect, have not been investigated. The aim of the present study was to investigate the effects of auraptene on melanogenesis in B16F10 murine melanoma cells.

## 2. Materials and Methods

### 2.1. Cell Culture

B16F10 murine melanoma cell line was purchased from Korean Cell Line Bank (Seoul, Korea). The cells were maintained at sub-confluence in a 95% air and 5% $CO_2$ humidified atmosphere at 37 °C. The medium used for the routine subculture was Dulbecco's Modified Eagle's Medium (DMEM) containing 10% fetal bovine serum (FBS), penicillin (100 units/mL), and streptomycin (100 µg/mL).

### 2.2. MTT Assay

Cell viability was determined using the 3-(4,5-dimethylthiazol-2-yl)-5-diphenyltetrazolium bromide (MTT) assay. B16F10 murine melanoma cells were cultured in 24-well plates for 18 h, followed by treatment with various concentrations (25, 50, and 100 µM) of auraptene for 48 h. In brief, MTT was added to the cells and the formazan crystals were dissolved in dimethyl sulfoxide. The absorbance was measured at 540 nm. The percentage of viable cells was determined relative to the control group.

### 2.3. Measurement of Melanin Content

The amount of melanin in B16F10 murine melanoma cells was measured according to a previously published method with slight modifications [17]. The cells were treated with auraptene and α-melanocyte-stimulating hormone (α-MSH) for 48 h at 37 °C. After removing the old media, cells were washed with cold phosphate-buffered saline (PBS) and cell pellets were dissolved in 1 N NaOH for 1 h at 80 °C. Spectrophotometric analysis of melanin content was performed at 405 nm absorbance. Each experiment was performed in triplicate.

### 2.4. Intercellular Tyrosinase Activity

Intercellular tyrosinase inhibitory activity was determined using a method previously described with slight modifications [18]. Arbutin was used as the positive control. In brief, B16F10 murine melanoma cells ($1.0 \times 10^5$) were seeded in a 60 mm dish and incubated at 37 °C in a humidified atmosphere with 5% $CO_2$ for three days. The cells were lysed with phosphate buffer containing 1% Triton X-100. The lysates were clarified by centrifugation for 15 min at 13,000 rpm.

Protein concentration of the lysate was determined and adjusted to the predetermined level using the lysis buffer. The lysate and different concentrations of the test samples were loaded into a 96-well plate, followed by the addition of 15 mM of L-DOPA into each well. After incubation at 37 °C for 1 h, absorbance at 475 nm was measured using an ELISA reader.

*2.5. Western Blotting Analysis*

After washing with cold PBS twice, B16F10 murine melanoma cells stimulated with α-MSH for 48 h were lysed with lysis buffer containing RIPA buffer, 1% Nonidet P-40, and 1% protease inhibitor cocktail for 1 h. The lysates were centrifuged at 15,000 rpm for 15 min at 4°C, and the supernatants were transferred to new microtubes. Protein concentration of the supernatant was determined with Bradford reagent (Bio-Rad) using bovine serum albumin (BSA) as the standard. After heating at 70 °C for 10 min, an equal amount of protein lysates were separated by electrophoresis on a 4–12% Bis-Tris mini gel (Invitrogen Inc., Gyeonggi-do, Korea), and the proteins were transferred to a nitrocellulose membrane (Invitrogen Inc. Gyeonggi-do, Korea). The membrane was then washed with Tris-buffered saline (TBS, 20 mM Tris base, 137 mM NaCl, pH 7.6) containing 0.1% Tween 20 (TTBS) and blocked in TTBS containing 5% skim milk for 24 h. The membrane was incubated overnight at 4 °C with TTBS-diluted (1:1000) primary antibodies (TRP-1, TRP-2, and tyrosinase) obtained from Santa Cruz Inc. The membrane was washed four times with TTBS before and after incubating for 1 h with a secondary peroxidase-conjugated goat immunoglobulin G (IgG) antibody (1:5000). Immunoreactive bands were visualized using an enhanced chemiluminescence solution followed by exposure to an X-ray film. The X-ray film was scanned to quantify intensity of bands.

*2.6. Data Analysis*

All data were expressed as means ± standard deviations of at least replicates. Student's *t*-tests and one-way analysis of variance were used for statistical analyses, and differences with *P* values of less than 0.05 were considered significant.

## 3. Results

*3.1. Auraptene Does Not Affect the Viability of B16F10 Murine Melanoma Cells*

The cytotoxicity of auraptene was tested on B16F10 murine melanoma cells at different concentrations (25–100 μM) for 48 h at 37 °C using the MTT assay. MTT is a yellow water-soluble tetrazolium salt. Metabolically active cells are able to convert the salt to a water-insoluble dark blue formazan by reductive cleavage of the tetrazolium ring. As shown in Figure 1, auraptene did not result in significant toxicity to B16F10 murine melanoma cells.

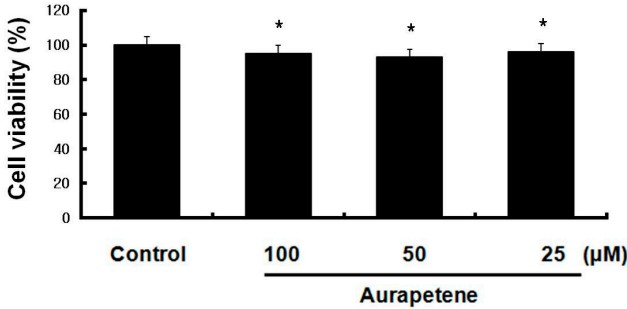

**Figure 1.** Cytotoxicity of auraptene on B16F10 murine melanoma cells by MTT assay. MTT assay was performed on B16F10 murine melanoma cells treated with different concentrations of auraptene for 48 h at 37 °C in a 5% $CO_2$ atmosphere. Absorbance was measured at 540 nm. The data represent the mean ± S.D of triplicate experiments. * $p < 0.05$, ** $p < 0.01$.

### 3.2. Auraptene Inhibits α-MSH-induced Melanin Synthesis and Intracellular Tyrosinase Activity in B16F10 Murine Melanoma Cells

To investigate the anti-melanogenic activity of auraptene, B16F10 melanoma cells were challenged with α-MSH to stimulate the production and release of melanin. As shown in Figure 2A, α-MSH-treated cells showed a marked increased the melanin content compared with untreated cells. Auraptene reduced melanin production with a half maximal inhibitory concentration ($IC_{50}$) value of 130 μM, while simultaneously causing 6.2% cell death. Moreover, melanin reducing effect of auraptene was a better result than arbutin, a positive whitening agent, which exhibited 12.2% melanin reducing effect at the 100 μM (Figure 2A). The amount of melanin content is suppressed to the previous results, indicating that it is associated with the activity of the enzyme involved in melanin synthesis.

Because melanin synthesis is regulated by tyrosinase, the direct inhibitory effect of auraptene on tyrosinase activity in B16F10 murine melanoma cells was examined using the intracellular tyrosinase assay. As shown in Figure 2B, auraptene decreased the activity of intracellular tyrosinase activity in a concentration-dependent manner.

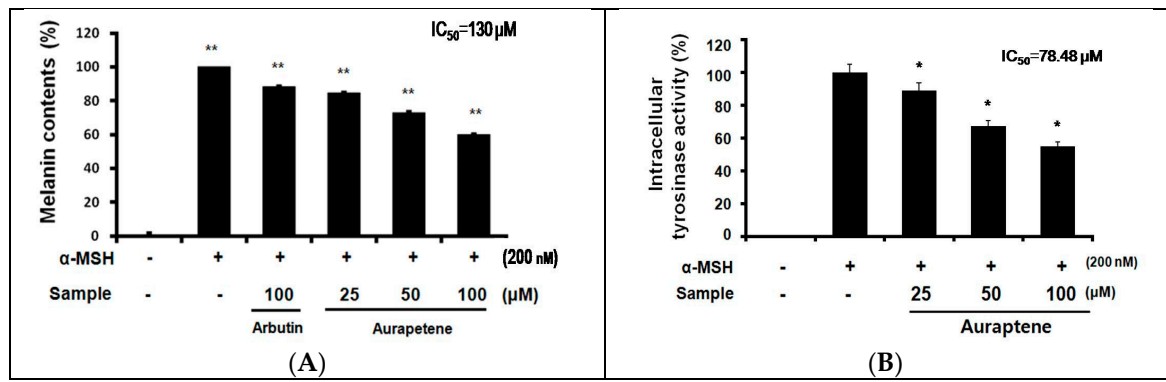

**Figure 2.** Auraptene inhibits α-MSH-induced melanogenesis and intracellular tyrosinase activity in B16F10 murine melanoma cells. B16F10 murine melanoma cells ($2.0 \times 10^4$ cells/mL) were pre-incubated for 18 h, melanin content, and intracellular tyrosinase activity were measured in the cells treated with α-MSH (200 nM) and auraptene after 72 h incubation at 37 °C in a 5% $CO_2$ atmosphere. Melanin contents (**A**) and intracellular tyrosinase activity (**B**) were measured as described in the 'Materials and Methods' section. The data represent the mean ± S.D of triplicate experiments. * $p < 0.05$, ** $p < 0.01$.

### 3.3. Auraptene Inhibits Melanogenesis-Related Proteins in B16F10 Murine Melanoma Cells

Western blotting analysis was performed to determine the expression of melanogenesis-related proteins such as tyrosinase, TRP-1, TRP-2, and microphthalmia-associated transcription factor (MITF). As shown in Figure 3, B16F10 murine melanoma cells treated with auraptene showed a significantly decreased expression of protein tyrosinase and TRP-2. However, TRP-1 protein expression was not affected. To understand the transcriptional regulation of tyrosinase and TRP-2 proteins, we investigated the effect of auraptene on the expression of MITF. Auraptene dose-dependently decreased the α-MSH-induced upregulation of MITF protein levels.

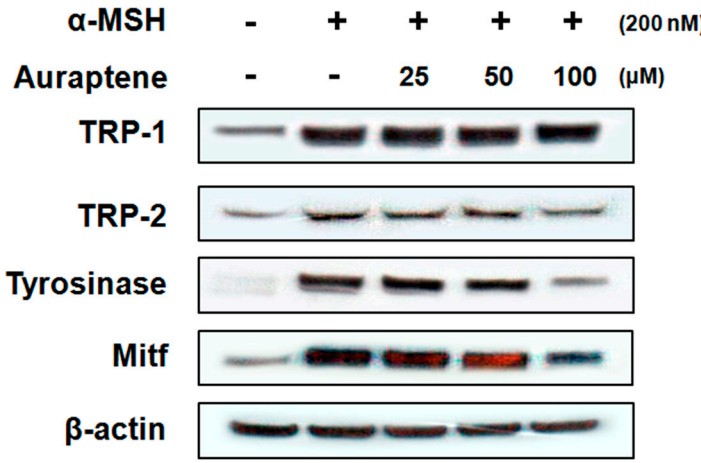

**Figure 3.** Effect of auraptene on protein levels of TRP-1, TRP-2, tyrosinase, and MITF in B16F10 murine melanoma cells. B16F10 murine melanoma cells ($1.0 \times 10^5$ cells/mL) were pre-incubated for 18 h, and the cells were stimulated with α-MSH (200 nM) in the presence of auraptene (25, 50, and 100 μM) for 72 h. Protein levels were determined by immunoblotting.

*3.4. Auraptene Stimulates ERK Phosphorylation*

Extracellular signal-regulated kinase (ERK) activation has been reported to inhibit melanogenesis [19,20]. Thus, we next investigated the hypopigmentary effect of auraptene on the activation of ERK. As shown Figure 4, auraptene induced phosphorylation of ERK in a dose-dependent manner. Furthermore, PD98059, a specific ERK inhibitor, attenuated auraptene-induced ERK phosphorylation at the highest concentration.

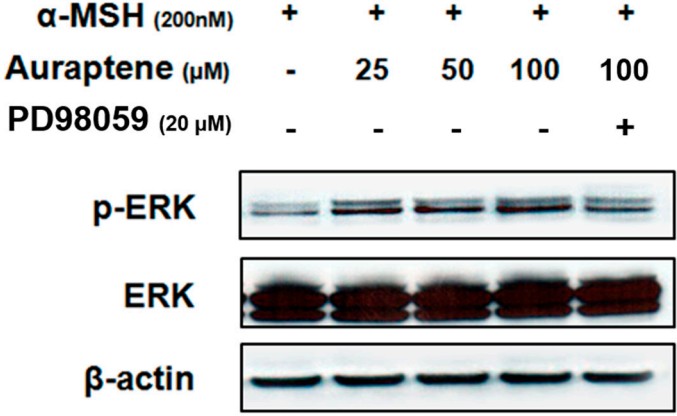

**Figure 4.** Auraptene increases phosphorylation of ERK in B16F10 murine melanoma cells. B16F10 murine melanoma cells ($1.0 \times 10^5$ cells/mL) were pre-incubated for 18 h and stimulated with α-MSH (200 nM) in the presence of auraptene (25, 50, and 100 μM) and PD98059 (20 μM) for 24 h. Protein levels were determined by immunoblotting.

## 4. Discussion

Citrus fruits contain bioactive compounds that have many beneficial aspects such as antihypertentive, anti-cardiac, and anticarcinogenic effects [21]. Auraptene, first isolated from members of the genus Citrus, is a natural bioactive monoterpene coumarin ether. Previous studies show that tyrosinase, a marker of melanin production, is inhibited by coumarin derivatives which suggest that these compounds can be used as the improvement agents for acquired hyperpigmentation

disorders [22–26]. However, the effect of auraptene on skin-related cells, especially the molecular aspect of its anti-melanogenesis effect, is limited.

First, we used the MTT assay to study the cytotoxicity of auraptene on B16F10 murine melanoma cells. Based on the MTT assay, between 80% and 90% of the cells remained viable after exposure to auraptene at its highest concentration, i.e., 100 μM for 48 h. These results suggest that the anti-melanogenetic activity of auraptene on B16F10 murine melanoma cells in the subsequent studies is not due to its cytotoxicity effect. Next, the inhibitory actions of auraptene on melanogenesis were evaluated.

Melanocytes can be stimulated by many effectors, including UV radiation and α-MSH. In this study, we used α-MSH (200 nM) to stimulate melanogenesis in B16F10 murine melanoma cells. As shown in Figure 2, B16F10 murine melanoma cells ($2.0 \times 10^4$) were pre-incubated for 18 h before use. The cells were treated with α-MSH (200 nM), arbutin (100 μM), and auraptene for 72 h at 37 °C in a 5% $CO_2$ atmosphere, and the amount of melanin secreted into the medium was measured. α-MSH-treated cells showed a significant increase in melanin production compared with untreated cells. Auraptene dose-dependently inhibited α-MSH-induced melanin production in B16F10 murine melanoma cells. At the highest concentration—i.e., 100 μM—auraptene inhibited 52.5% of melanin biosynthesis.

Tyrosinase is a key enzyme in the biosynthesis of melanin. Therefore, inhibition of tyrosinase is a major strategy in the development of new whitening agents. The effects of auraptene on the catalytic activities of tyrosinase are shown in Figure 2B. The inhibitory effects of auraptene on tyrosinase activities were reflected in the reduced amount of melanin synthesized. It should be noted that the reduced melanin contents were attributed to the suppression of the tyrosinase activity.

Melanogenesis is known to be regulated by a complex cascade of enzymes, such as tyrosinase, TRP-1, TRP-2, and MITF [27]. In particular, MITF is the major transcriptional regulator for genes of melanogenic enzymes. Upstream of the signaling cascade, α-MSH binds to its receptor for increasing cyclic adenine monophosphate (cAMP) levels and activating cAMP-dependent protein kinase. This could result in the phosphorylation of cAMP response element-binding protein and an increase in MITF protein levels. Activated MITF increases the expression of tyrosinase and other melanogenic enzymes, which in turn leads to an increase in melanin synthesis. Therefore, the downregulation of melanogenic enzymes in melanin synthesis offers a new strategy for skin whitening [28].

To understand the molecular mechanisms underlying the reduction of melanogenesis by auraptene, we measured the expression of melanogenesis-related proteins, such as tyrosinase, TRP-1, and TRP-2. As shown in Figure 3, auraptene dose-dependently reduced the protein levels of tyrosinase and TRP-2. However, TRP-1 was not affected by auraptene treatment. In addition, MITF, the transcriptional regulator of tyrosinase and TRP-2, decreased in the presence of auraptene in a dose-dependent manner. These results suggest that auraptene inhibits the expression of tyrosinase and TRP-2 proteins through the downregulation of MITF.

Several reports have demonstrated that ERK activation led to the phosphorylation of MITF and its subsequent ubiquitination and degradation [20]. Because auraptene was found to inhibit melanogenesis, we investigated whether the inhibitory effects were associated with ERK activation by western blotting. As shown in Figure 4, auraptene induced phosphorylation of ERK in a dose-dependent manner. Moreover, PD98059, a specific inhibitor of ERK, suppressed the phosphorylation of ERK induced by auraptene, suggesting that auraptene inhibits melanogenesis via ERK signaling pathways.

In conclusion, auraptene inhibited α-MSH-induced cellular melanin biosynthesis and tyrosinase activity in B16F10 murine melanoma cells by reducing the protein level of MITF, tyrosinase, and TRP-2, and increasing ERK activity. These results suggest that auraptene inhibits melanogenesis by activating the ERK pathway-mediated suppression of MITF and its downstream target gene, especially tyrosinase. Our results suggest that auraptene is useful in the development of skin-whitening agents.

**Acknowledgments:** This work was carried out with the support of "Cooperative Research Program for Agriculture Science & Technology Development (Project No. PJ010934082015)" Rural Development Administration, Republic of Korea.

**Author Contributions:** Chang-Gu Hyun conceived and designed the experiments; Min-Jin Kim and Sang Suk Kim performed the experiments; Young Hun Choi and Nam Ho Lee analyzed the data; Kyung-Jin Park and Hyun Joo An contributed reagents/materials/analysis tools; Chang-Gu Hyun wrote the paper.

**Conflicts of Interest:** The authors declare no conflict of interest.

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
