# Peer review of "Anti-melanogenic Activity of Auraptene via ERK-mediated MITF Downregulation"

_cosmetics, doi:10.3390/cosmetics4030034_

Round 1

Reviewer 1 Report

@font-face {   font-family: "Cambria Math"; }@font-face {   font-family: "Yu Mincho"; }@font-face {   font-family: "@Yu Mincho"; }@font-face {   font-family: "MS Gothic"; }@font-face {   font-family: "@MS Gothic"; }p.MsoNormal, li.MsoNormal, div.MsoNormal { margin: 0mm 0mm 0; text-align:

The manuscript entitled “Anti-melanogenic activity of auraptene via

3 ERK-mediated MITF downregulation” by  Min-Jin Kim , Sang Suk Kim , Kyung-Jin Park , Hyun Joo An , Young Hun Choi , Nam Ho Lee ,

 and Chang-Gu Hyun examined the anti-melanogenic effect of auraptene by using some biological assays.

First of all, the authors should provide the information of the chemical structure of auraptene. It is quite important from the structure-activity relationship.

Aurapten is the family of coumarin derivatives bearing isoprenoid side chains.

However, the authors did not argue these points, the authors showed only the results. There are many similar works dealing with this kind of topic, the authors should honestly describe or quote previous works.

The biological works done by authors are not updated molecular biological works, all works done by authors are qualitative, not quantitative.

So, it is better to carry out more precise quantitative works.

Based on these facts, the manuscript should be revised (added some more experimental results)

Author Response

<Cosmetics>

< Anti-melanogenic activity of auraptene via ERK-mediated MITF downregulation>

Dear Editor,

Thank you for your useful comments and suggestions on the structure of our manuscript. We have revised the WHOLE manuscript carefully and tried to avoid any grammar or syntax error. In addition, we brought a submitted paper to a native speaker of English (Enago ; http:// www.enago.com/). We have modified the manuscript accordingly, and detailed corrections are listed below point by point:

Editor # 1

1. Why in Fig 1 neither standard deviation nor significancy are reported?.

- As shown in Fig. 1, we have checked it. (P3L135)

2. Fig 3 e 4 are identical.. probably an error was made.

- We are so sorry. We have revised the Fig 4. (P5L178)
3. In introduction, the authors should report that plant molecules are normally synthetized for a specifi protective function in plants and that they can also act on human cells, modifying their metabolism.. ie. antoxidant activity, ability to activate specific genes… I suggest to cite the following work to improve this section: Biochemical Journal, 2001, 354(3), 493-500; Vitis 2017, 56, 19–26; Journal of medicinal plants research, 2011, 5(31), 6697-6703 and other chosen by the authors.

- We have added and rearranged the references. (P2L55-58/P7L270-276)

4. English could be revised.

- We brought a submitted paper to a native speaker of English (Enago ; http:// www.enago.com/).

Editor # 2

1. First of all, the authors should provide the information of the chemical structure of auraptene. It is quite important from the structure-activity relationship. Aurapten is the family of coumarin derivatives bearing isoprenoid side chains. However, the authors did not argue these points, the authors showed only the results. There are many similar works dealing with this kind of topic, the authors should honestly describe or quote previous works.;

- We totally agree with editor’s opinion. The previous works of the family of coumarin derivatives have been revised the section of “Discussion”. (Please see page 5/line 187 to page 6/line 190)

- We have added 5 references rearranged them. (P8295-306)

2 The biological works done by authors are not updated molecular biological works, all works done by authors are qualitative, not quantitative. So, it is better to carry out more precise quantitative works. Based on these facts, the manuscript should be revised (added some more experimental results);

- As shown in Fig 2A and B, we have addressed IC50 values as quantitative numbers. Concerning as Fig 3, we would like to say that our lab still do not have the Gel Doc program to evaluate quantitative numbers. At present, we are totally dependent on your generous mind.

Reviewer 2 Report

The paper entitled “Anti-melanogenic activity of auraptene via ERK-mediated MITF downregulation” is a good work which reports the capacity of a plant molecule, the auraptene, to inhibit melanin production. However, I have some minor revisions to improve the paper:

a) why in Fig 1 neither standard deviation nor significancy are reported?

b) fig 3 e 4 are identical.. probably an error was made.

c) in introduction, the authors should report that plant molecules are normally synthetized for a specifi protective function in plants and that they can also act on human cells, modifying their metabolism.. ie. antoxidant activity, ability to activate specific genes… I suggest to cite the following work to improve this section: Biochemical Journal2001, 354(3), 493-500; Vitis 2017, 56, 19–26; Journal of medicinal plants research2011, 5(31), 6697-6703 and other chosen by the authors.

d) English could be revised.

Author Response

(The authors gave the same response as above.)

Round 2

Reviewer 1 Report

The authors revised the manuscript according to the referee's suggestions. They added some references in the manuscript, however, they did not add any important experiments. In recent years, it is common to evaluated the strength of the protein expression. The quantitative analysis based on the PCR method is widely used.

If the authors do not (ca not) use this methodology, the authors should provide the another important information, however, the reviewer can not find such information in the manuscript.

Author Response

Editor # 1

1. The authors revised the manuscript according to the referee's suggestions. They added some references in the manuscript, however, they did not add any important experiments. In recent years, it is common to evaluated the strength of the protein expression. The quantitative analysis based on the PCR method is widely used.

- Thank you for your useful comments and suggestions on the structure of our manuscript. We also agree entirely with your opinion. On the other hand, we would like to ask you to reconsider your request about further study (RT-PCR). I'd like to make a little bit of excuses. At present, many articles do not necessarily require RT-PCR studies. Please see the attachment below. Nonetheless, if you require RT-PCR data, we need about three months more time. We are totally dependent on your generous mind.

[Int. J. Mol. Sci. 2017, 18] Plumbagin Suppresses –MSH-Induced Melanogenesis in B16F10 Mouse Melanoma Cells by Inhibiting Tyrosinase Activity

[Molecules 2016, 21] Constituents of Cryptotaenia japonica Inhibit Melanogenesis via CREB- and MAPK-Associated Signaling Pathways in Murine B16 Melanoma Cells

[Int. J. Mol. Sci. 2016, 17] Anti-Melanogenic Activities of Heracleum moellendorffii via ERK1/2-Mediated MITF Downregulation

[Molecules 2014, 19, 12940-12948] Effect of Chlorogenic Acid on Melanogenesis of B16 Melanoma Cells

[Chemico-Biological Interactions 245 (2016) 66-71] Chaetocin inhibits IBMX-induced melanogenesis in B16F10 mouse melanoma cells through activation of ERK

[Biochemical and Biophysical Research Communications 480 (2016) 648-654] Inhibition of tyrosinase activity and melanin production by the chalcone derivative 1-(2-cyclohexylmethoxy-6-hydroxy-phenyl)-3-(4-hydroxymethyl-phenyl)-propenone

[Life Sciences 143 (2015) 43–49] Isosakuranetin, a 4′-O-methylated flavonoid, stimulates melanogenesis in B16BL6 murine melanoma cells

2. If the authors do not (ca not) use this methodology, the authors should provide the another important information, however, the reviewer can not find such information in the manuscript.

- The academic significance of this paper can be summarized in two ways : The first is that whitening effect of auraptene is more effective than arbutin (a commercial agent), and the second is that it is involevd in an ERK signaling pathway. We believe that these findings will provide useful information to readers of Cosmetics Journal. In relation to this, the paper has been revised and emphasized. (Line 145 to 148, Line 175 to 180, Line 236 to 241)

Round 3

Reviewer 1 Report

The authors revised the manuscript, however, the revision is not well designed. As pointed out in the previous report, the reviewer pointed out the following.

In recent years, it is common to evaluated the strength of the protein expression. The quantitative analysis based on the PCR method is widely used. If the authors do not (can not) use this methodology, the authors should provide  another important information, however, the reviewer can not find such information in the manuscript.

The situation is the same as the previous version. If the authors insist on the concentration dependent, it is required to endorse it. The qualitative analysis is not enough to support the conclusion.

Cosmetics EISSN 2079-9284 Published by MDPI AG, Basel, Switzerland RSS E-Mail Table of Contents Alert
Back to Top